# Substrate scope expansion of 4-phenol oxidases by rational enzyme selection and sequence-function relations

Daniel Eggerichs ⓘ , Nils Weindorf, Heiner G. Weddeling ⓘ , Inja M. Van der Linden & Dirk Tischler ⓘ ✉

Enzymes are natures' catalysts and will have a lasting impact on (organic) synthesis as they possess unchallenged regio- and stereo selectivity. On the downside, this high selectivity limits enzymes' substrate range and hampers their universal application. Therefore, substrate scope expansion of enzyme families by either modification of known biocatalysts or identification of new members is a key challenge in enzyme-driven catalysis. Here, we present a streamlined approach to rationally select enzymes with proposed functionalities from the ever-increasing amount of available sequence data. In a case study on 4-phenol oxidoreductases, eight enzymes of the oxidase branch were selected from 292 sequences on basis of the properties of first shell residues of the catalytic pocket, guided by the computational tool A²CA. Correlations between these residues and enzyme activity yielded robust sequence-function relations, which were exploited by site-saturation mutagenesis. Application of a peroxidase-independent oxidase screening resulted in 16 active enzyme variants which were up to 90-times more active than respective wildtype enzymes and up to 6-times more active than the best performing natural variants. The results were supported by kinetic experiments and structural models. The newly introduced amino acids confirmed the correlation studies which overall highlights the successful logic of the presented approach.

The exponentially growing number of sequences in public databases reveals more and more of nature's treasure chest[1–3]. At the same time, handling big data with ten-thousands of protein candidates becomes increasingly important to select suitable biocatalysts, as enzymatic approaches are at the forefront of synthetic applications. This development is reasoned by the excellent selectivity of enzymes and their ability to activate structural motifs, which were difficult, if not impossible, to target by classical chemical methods[4–6]. Therefore, efficient discovery and improvement of biocatalysts is an emerging topic[7–9]. Despite excellent tools for enzyme improvement, like directed evolution, a good and well-chosen starting point for a muta-genesis campaign saves resources and time[10–12]. Thus, computational approaches gain popularity for enzyme discovery[7,13]. Nowadays, many approaches, like sequence similarity networks (SSNs), allow the clustering of a vast sequence space[14,15]. But considering that single mutations can already change the substrate scope or the stability of enzymes dramatically, it becomes apparent that a more focused investigation of a few hundred sequences is beneficial to fetch more modest differences between the cata-lysts. Thus, the here-described approach considers the diversity of the first shell residues of the catalytic center, which requires a certain knowledge of

the enzyme family or structural information as a starting point for residue selection. However, to great advancements in the field of structure predic-tion, structural models are nowadays remarkably easy to obtain, e.g., by using the AlphaFold2 algorithm[16,17].

To demonstrate the applicability and the potential of this approach, the family of flavin-dependent 4-phenol oxidoreductases was chosen, which was recently investigated for its potential in the utilization of lignin-derived compounds[18,19]. The family is part of the widespread vanillyl alcohol oxi-dase/p-cresol methyl hydroxylase (VAO/PCMH) superfamily, among which the family of 4-phenol oxidoreductases distinguishes itself from other families as it is comprised of dehydrogenases and oxidases, which are further divided into bacterial and fungal enzymes[20,21]. This diversity within the family serves as an excellent model system as it allows for unique sequence features to be compared between all enzyme groups.

Within the 4-phenol oxidoreductase family, two fungal and seven bacterial sequences were described until today. The fungal vanillyl alcohol oxidases (VAOs) from *Diplodia corticola* and *Penicillium simplicissimum* share a similar substrate scope with differences for substitution patterns at the aromatic ring in *o*-position[22–24]. Of the bacterial enzymes, three oxidases

Microbial Biotechnology, Ruhr University Bochum, Universitätsstr. 150, 44780 Bochum, Germany.
✉e-mail: dirk.tischler@rub.de

and four dehydrogenases are described. The eugenol oxidases (EUGOs) from *Rhodococcus jostii* RHA1 and *Nocardioides* sp. YR527, and the 4-ethyl phenol oxidase (4EPO) from *Gulosibacter chuangengensis* represent the oxidase branch[25–27], while the *p*-cresol methyl hydroxylase (PCMH) from *Pseudomonas putida*, the eugenol hydroxylase (EUGH) from *Pseudomonas* sp. OPS1, and the pinoresinol-α-hydroxylases (PRαHs) from *Burkholderia* sp. SG-MS1, as well as *Pseudomonas* sp. SG-MS2 are members of the dehydrogenase family[28–30]. While the oxidases use dioxygen as a terminal electron acceptor, the dehydrogenases are cytochrome c dependent[31]. All enzymes harbor a covalently bound flavin adenine dinucleotide (FAD) cofactor as a prosthetic group and accept phenolic substrates with varying substituents in *p*-and *o*-position to the hydroxy group. The size of the accepted substrate molecules ranges from small compounds like 4-cresol to the bulky tetrahydrofuran lignan pinoresinol. Overall, a broad reaction spectrum is observed, which includes hydroxylations in 4α- and 4γ-position, dehydrogenation, oxidative deamination, and cleavage of benzylic ethers (Fig. 1)[23,25,32]. Mechanistic studies for the VAO from *P. simplicissimum* (*Ps*VAO) revealed that a hydride is transferred from the benzylic position of the substrate to the N5 atom of the FAD cofactor which results in the formation of a methide intermediate[33]. From this intermediate, either a proton is abstracted or water attacks as a nucleophile to yield the oxidized product. The catalytic cycle is closed with a two-electron transfer to the respective electron acceptor.

The high diversity in catalyzed reactions and substrate scope makes the enzyme family of 4-phenol oxidoreductases an interesting case study. Stereo- and regioselective oxidation is a cornerstone of (organic) synthesis, and phenolic compounds represent a common drug motive. Therefore, we decided to expand nature's toolbox for these reactions while providing a streamlined approach for rational enzyme selection which includes a general-use software tool for sequence analysis (A²CA)[34]. Within this work, we demonstrated the capabilities of A²CA as a user-friendly, sequence-based alignment tool that allows for quick visualization and setting of selection criteria for efficient exploration of the natural sequence space. From the initial analysis, bacterial 4-phenol oxidases emerged as the most diverse branch of the family and were subsequently studied in detail. Within this exceptional versatile enzyme class, eight enzymes were selected by A²CA guidance and robust sequence-function relations were established by

correlations of the residues' diversity with the enzyme activity. In combination with an efficient oxidase screening assay, directed evolution of identified hot spot residues allowed us to expand the natural sequence space of 4-phenol oxidases towards substrates with non-natural substituents in *o*- and *p*-position.

## Results

### A²CA-guided enzyme selection based on function-specific clustering of the catalytic center

To streamline the analysis of the family of 4-phenol oxidoreductases, the first-shell amino acid residues of the catalytic pocket were grouped into five functional clusters according to their characteristics, which were derived from literature and geometric considerations (Fig. 2a). To account for residue movement, the EUGO from *Rhodococcus jostii* RHA1 (*Rj*EUGO), was chosen as template, since five crystal structures were available[35]. Cluster properties resulted in the following four hypotheses regarding substrate binding: H1: Based on earlier mutagenesis studies, it can be speculated that the residues of the P-cluster are essential for substrate binding[36]. H2: Residues of the T-cluster and H-cluster likely interact with the substrates' *o*-substituent(s). H3: The polar W-cluster probably interacts with the water nucleophile or polar groups of the substrate itself and, thus, is decisive for the reaction type. H4: The hydrophobic A-cluster likely restricts the size of the *p*-substituent.

Database searches resulted in 292 unique sequences, which clustered in three major clades containing subclades with characterized enzymes (Fig. 3a). Using the software tool A²CA, the first shell residues of the catalytic center were highlighted to display the natural cluster variability among the enzyme family (Fig. 3b). In agreement with H1, the P-cluster was found to be conserved for the all 4-phenol oxidoreductases. Regarding other clusters, large subclade-specific differences in diversity were observed. While PCMHs and EUGHs sequences contain little changes in the amino acid composition, diverse patterns were obtained for PRαHs, EUGOs, and 4EPOs, of which the bacterial 4-phenol oxidase branch (EUGOs and 4EPOs) was selected for detailed analysis. Next to the high diversity on the sequence level, oxidases require no co-substrate except readily available dioxygen and represent, therefore, excellent model systems.

**Fig. 1 | Reaction schema of 4-phenol oxidoreductases.** A hydride from the 4α-position is transferred to the FAD cofactor, which results in the formation of a quinone-like methide intermediate. In dependency of group **X** and the particular enzyme, different products are formed, which fall in one of the four shown categories (black boxes).

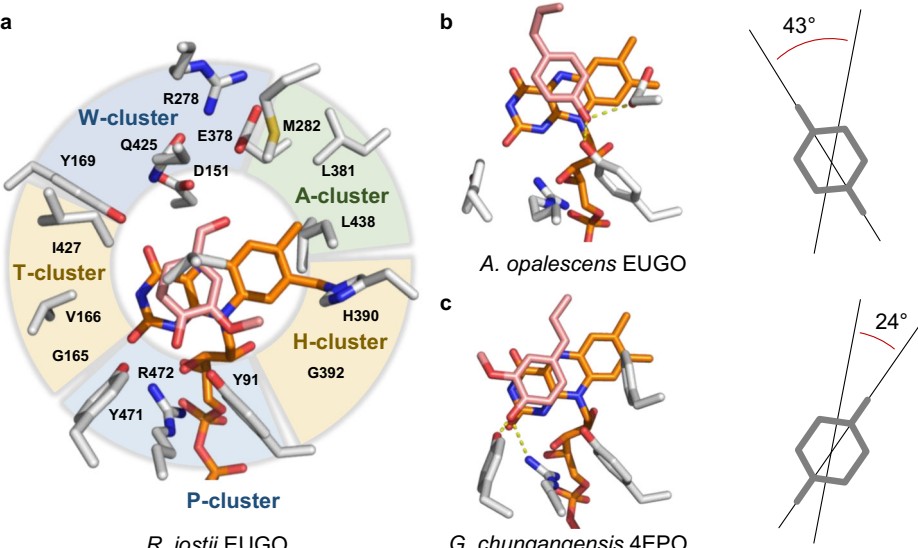

**Fig. 2 | Substrate binding in 4-phenol oxidases. a** Functional cluster within the first-shell residues of the catalytic center in the crystal structure of *Rj*EUGO (PDB: 5FXP). The FAD cofactor is shown in orange, while the substrate vanillyl alcohol (**5**) is shown in light red. The P-cluster is comprised of Y91, Y471, and R472 which are essential for the binding of the phenolate moiety of the substrate. The T-cluster is located at the substrate entrance tunnel, and residues G165, V166, and I427 may interact with an *o*-substituent of the bound substrate molecule. Likewise, G392 and H390 in the H-cluster are in contact with the other *o*-substituent. The substrate's *p*- substituent interacts with the polar residues D151, Y169, R278, E378, and Q425 of the W-cluster on one side, while it is in contact with the non-polar residues M282, L381, and L438 of the A-cluster on the other side. V436 is located on top of the substrate. **b** Substrate orientation in the homology model of EUGO from *Allonocardiopsis opalescens* (*Ao*EUGO) after 25 ns of simulation. The rotation relative to the 1,4-axis in *Rj*EUGO is shown. **c** Substrate orientation in the crystal structure of 4EPO from *Gulosibacter chungangensis* (*Gc*4EPO, PDB: 7BPI).

Based on the derived hypotheses (H2 to H4), the oxidases were studied by A²CA with regard to residue size in the A-, T-, and H-cluster and polarity in the W-cluster. The oxidases from *Streptomyces cavernae* (*Sc*EUGO) and *Geodermatophilus sabuli* (*Gs*EUGO) were identified with comparably small T-cluster residues (Fig. S1). On the other end of the spectrum, the enzyme from *Gulosibacter chungangensis* (*Gc*4EPO) was found to contain sterically demanding residues in the T-cluster, matching the recently described narrow substrate pocket[26]. As oxidases with mid-sized catalytic pockets, the enzymes from *R. jostii* (*Rj*EUGO), *Geodermatophilaceae bacterium* (*Gb*EUGO), and *Norcadioides* sp. (*Nsp*EUGO) were selected. The oxidases from *Allonocardiopsis opalescens* (*Ao*EUGO) and *Arthrobacter* sp. UCD-GKA (*Asp*EUGO) stood out in terms of polarity in the W- and H-cluster (Figs. S2 and S3). Further deviations from the consensus in the W-cluster were observed for *Gc*4EPO (Fig. S3), while no significant changes with respect to residue size or hydrophobicity were observed in the A-cluster (Fig. S4). In total, eight oxidases were selected for this study, of which five have not been described before (Table S1). The enzymes share a sequence identity of 76 to 50% (Table S2). All enzymes were designated as eugenol oxidases (EUGOs) with the exception of *Gc*4EPO for consistency with earlier studies. As *Rj*EUGO was used as a template, residue numbering refers to this sequence if not stated otherwise.

All enzymes were successfully expressed in *E. coli* (Figs. S5–S12) and were found to have comparably physical characteristics (Table S3, Figs. S13 and S14). All eight oxidases were tested for their activity on 46 compounds to collect sufficient data for structure-function relations (Table S4). Product formation was validated by GC-MS measurements (Table S5).

## Modulation of enzyme activity through substrate rotation by residue 392

Among the selected oxidases, residue 392 in the catalytic center was found to be remarkably variable (Fig. 3b): Five enzymes carry a Gly residue, while *Gc*4EPO contains a Phe, *Asp*EUGO a Ser, and *Ao*EUGO an Asp. Further, some oxidases from rhodococci harbor a Cys in this position but were not selected for this study due to their high similarity to *Rj*EUGO. This

naturally occurring diversity in residue size and polarity coincided with deviations from the expected substrate acceptance of these enzymes. Thus, we investigated the role of this position in detail. For *Gc*4EPO, a selectivity for non-methoxylated substrates would be expected due to the steric demand of Phe392. But no activity was found for chavicol (**1**) or 4-hydroxybenzyl alcohol (**4**), while an outstanding activity for 4-ethyl phenol (**32**) was observed ($4.58 \pm 0.18$ s⁻¹, Table S4). *Ao*EUGO, harboring a sterically demanding but polar Asp392, was also found incapable of converting **1**, while on the contrary, the highest activity for **4** was detected ($2.6 \pm 0.04$ s⁻¹). These drastic changes in substrate acceptance were found to be reasoned in a substrate rotation inside the catalytic pocket. In *Ao*EUGO, the phenolate group of **1** was coordinated by residues 471 and 392, after 25 ns of simulation, instead of the canonical triad, resulting in unfavorable steric interactions of the *p*-allyl substituent with the W-cluster (Fig. 2b, S15 and S16). In contrast, increased polar interactions for the *p*-hydroxy substituent in **4** are beneficial for substrate turnover. In the crystal structure of *Gc*4EPO, interactions of the *p*-substituent with the W-cluster are reduced as the substrate is rotated in the other direction (Fig. 2c), which is likely caused by the steric effect of Phe392. The greater distance towards the polar cluster is in agreement with the enzyme's low activities for reactions involving the addition of water and likely contributes to the observed favor for dehydrogenation reactions. The increased activity for these reactions becomes apparent for comparably high activities on **32** as well as vanillyl alcohol (**5**) and 3,4-dihydroxybenzyl alcohol (**7**, Table S4).

A substrate rotation could not be observed for *Asp*EUGO as Ser392 is neither influential enough from a steric nor from a polarity aspect (Fig. S17). Nevertheless, steric effects of the residue were identified to increase the affinity for non-methoxylated benzyl alcohols in a comparative study with *Gb*EUGO at optimal conditions for both enzymes (Figs. S18–S21). The $K_M$ value of $33 \pm 2$ µM for **4** was found to be 12 times lower than for *Gb*EUGO ($455 \pm 50$ µM), while similar $k_{cat}$ values around $18 \pm 1$ s⁻¹ were determined at pH 9.5 (Fig. S22, Table S6). A similar picture is indicated for 3,4-dihydroxybenzyl alcohol (**7**), but interference of the substrate with the assay made the collection of reliable data difficult (Fig. S23). In contrast, no

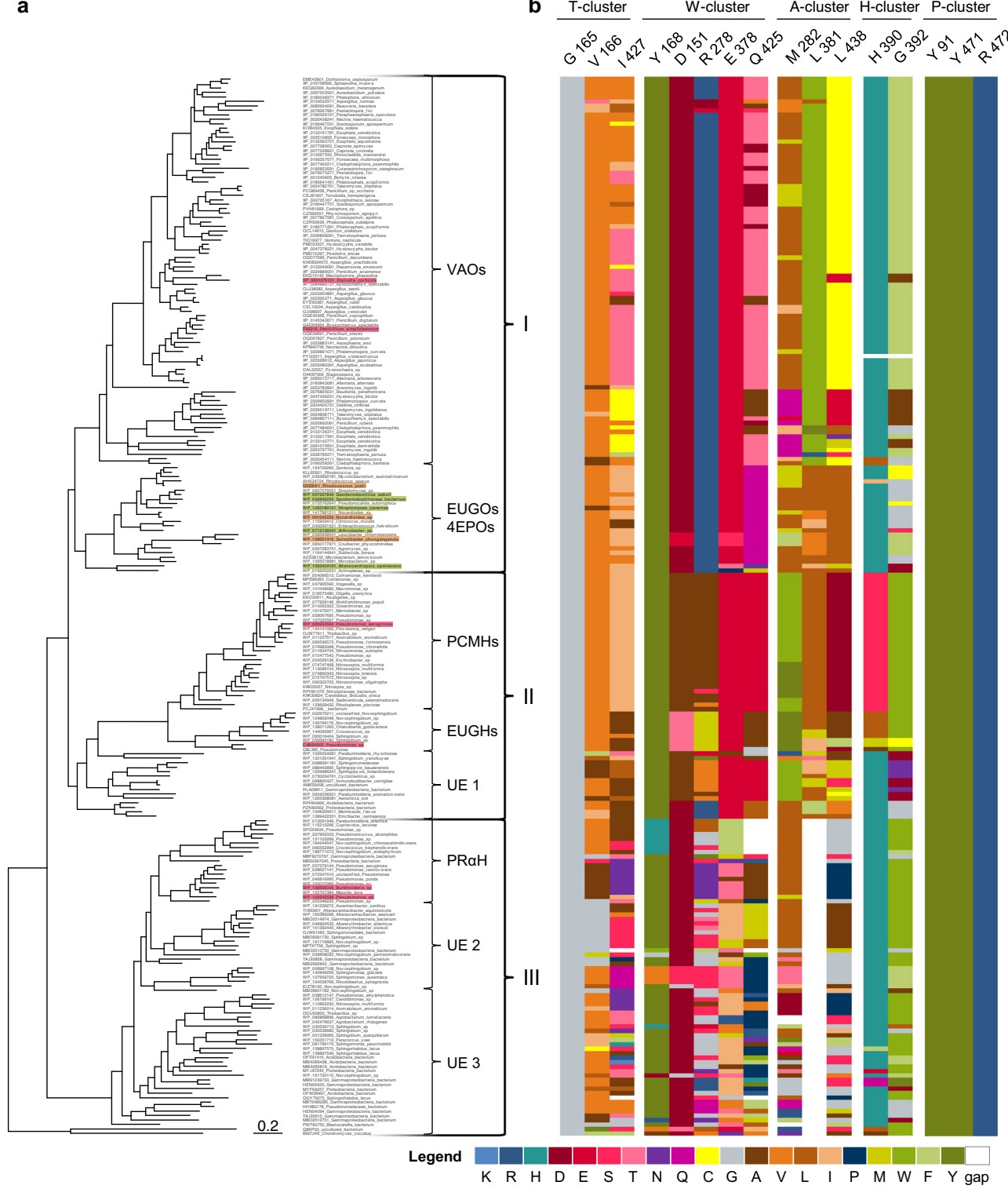

**Fig. 3 | A²CA analysis of the FAD-dependent 4-phenol oxidoreductase family.**
**a** The phylogenetic tree clusters in three major clades (I–III). The respective sub-clades are named according to characterized enzymes. VAOs vanillyl alcohol oxidases, EUGOs eugenol oxidases, 4EPOs 4-ethylphenol oxidases, PCMHs *p*-cresol methyl hydroxylases, EUGHs eugenol hydroxylases, PRαHs pinoresinol-α-hydroxylases, UE uncharacterized enzymes. Characterized enzymes (light red): VAOs from *Diplodia corticola* (XP_0201276121) and *Penicillium simplicissimum* (P56216), PCMH from *Pseudomonas aeruginosa* (WP_080263564) was not characterized but shares 91% sequence identity to the characterized enzyme from *Pseudomonas putida* (WP_032489501) which was filtered out by removal of redundancies, EUGH from *Pseudomonas* sp. OPS1 (AAM21269), PRαHs from

*Burkholderia* sp. SG-MS1 (WP_168666336) and *Pseudomonas* sp. SG-MS2 (WP_105642528). Characterized enzymes used in this study (light orange): EUGOs from *Rhodococcus jostii* RHA1 (Q0SBK1) and *Nocardioides* sp. YR527 (WP_091045259) and 4EPO from *Gulosibacter chungangensis* (WP_158051316). Enzymes characterized in this study (green): 4-phenol oxidases from *Geodermatophilus sabuli* (WP_097207849), *Geodermatophilaceae bacterium* (WP_026846239), *Streptomyces cavernae* (WP_1283780151), *Arthrobacter* sp. (WP_0712138341) and *Allonocardiopsis opalescens* (WP_1062434191). **b** A²CA results for first shell residues in the catalytic center. The residues are grouped according to the clusters in Fig. 2. The indicated positions correspond to the crystal structure of the EUGO from *R. jostii* (PDB: 5FXP).

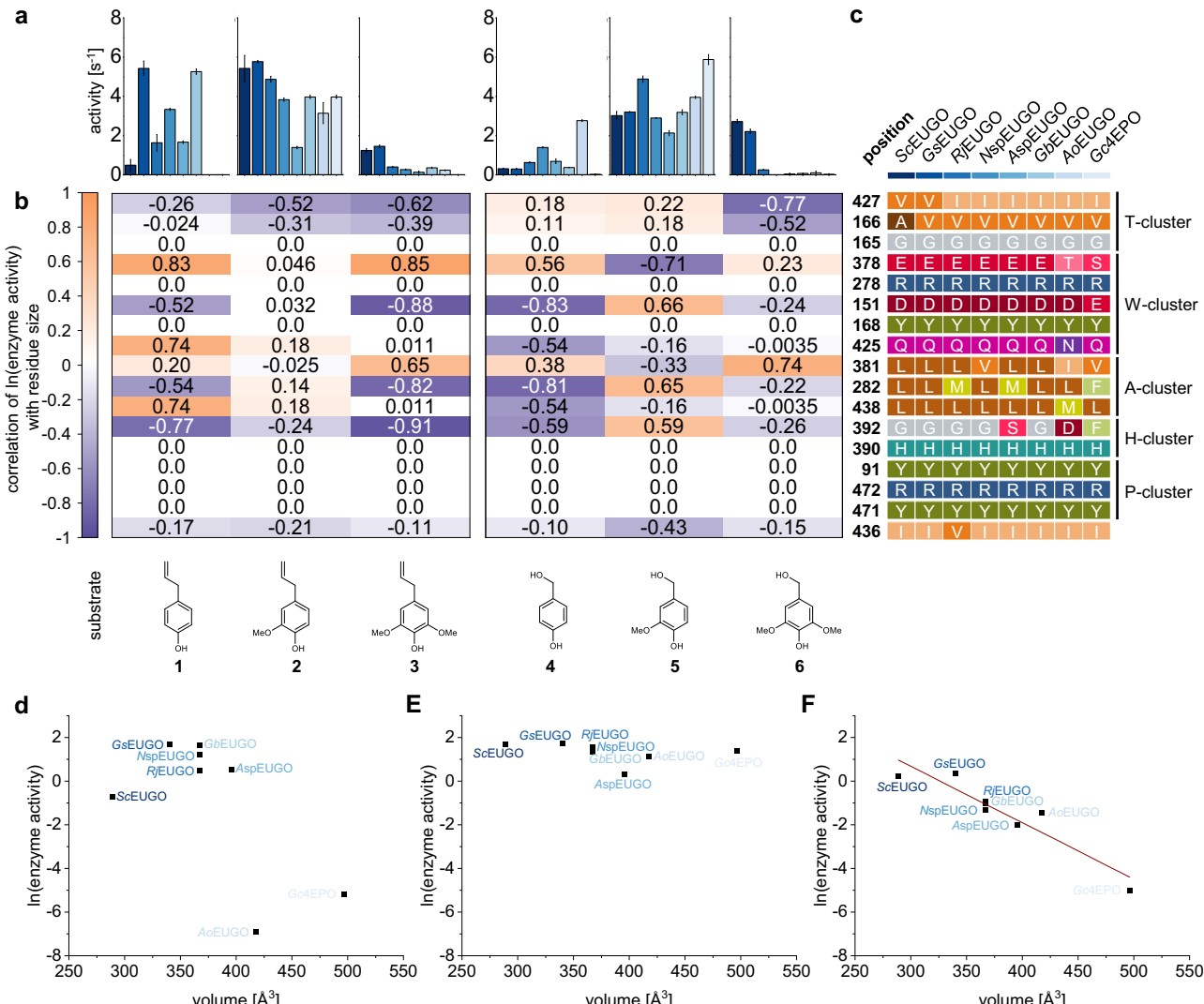

**Fig. 4 | First-shell residues of the catalytic pocket influencing the acceptance of substrates with varying *o*-methoxylation.** Compounds with *p*-allyl and -benzyl alcohol groups were compared to mitigate the influence of the *p*-position. **a** Observed activity on the indicated substrates by the color-coded oxidases of this study (compare Table S2). The error bars represent the standard deviation from a triplicate measurement. **b** Heatmaps for the correlation of the logarithmic enzyme activity with the change in the size of each individual residue among the selected oxidases. A negative correlation of the amino acid size and the activity means that for enzymes containing larger residues, lower activities were observed and vice versa. Conserved residues have no correlation. **c** Natural diversity of residues present at the indicated position of each oxidase. The oxidases are color-coded according to a, and the residues are grouped according to functional clusters in the catalytic center of *Rj*EUGO (see Fig. 2a). **d** Plot of the combined amino acid volume of positions 166, 392, and 427 against the logarithm of the observed enzyme activity for chavicol (**1**). **e** Plot of the combined amino acid volume of positions 166, 392, and 427 against the logarithm of the observed enzyme activity for eugenol (**2**). **f** Plot of the combined amino acid volume of positions 166, 392, and 427 against the logarithm of the observed enzyme activity for 4-allyl-2,6-dimethoxy phenol (**3**). A correlation between residue size and activity can be seen.

differences in $K_{\mathrm{M}}$ were observed for the *o*-methoxylated substrates eugenol (**2**) or vanillyl alcohol (**5**, Figs. S24 and S25).

### Sequence-function relations obtained from correlations between residue variety and enzyme activity

As the example of position 392 highlights the strong effect a single residue can have, the individual influence of all 17 residues of the first shell was investigated to verify the remaining hypotheses H2 to H4. The influence of the T- and H-cluster residues on the *o*-substitution pattern (H2) was investigated for 4-allyl phenols and 4-hydroxy benzyl alcohols (Fig. 4a). Therefore, the correlation between the logarithm of the activity and the residue size was calculated (Fig. 4b). Increasing activity with increasing residue size results in a positive correlation, while a negative correlation indicates the beneficial effect of small residues. For the T-cluster residues, an increasingly negative correlation with the number of *o*-methoxy groups is

observed, which highlights that smaller residues are required for the acceptance of di-*o*-methoxylated substrates. For residue 392 of the H-cluster, no clear pattern was observed due to the before-mentioned alternative substrate binding modes, which overcompensate steric effects. Thus, it can be concluded that T-cluster residues have the highest contribution for the selectivity towards di-*o*-methoxylated substrates, while the repositioning effects of the H-cluster residue 392 is a key factor for the conversion of non-methoxylated substrates. In agreement with this, *Ao*EUGO and *Gc*4EPO were identified as outliers for the conversion of chavicol (**1**, Fig. 4d), while all enzymes were active on eugenol (**2**), justifying the "EUGO" designation (Fig. 4e). In contrast, steric factors dominate the acceptance of 4-allyl-2,6-dimethoxy phenol (**3**), so that the influence of the T- and H-cluster is visible (Fig. 4f).

Next to the number of *o*-substituents, the acceptance of different chemical groups in *o*-position could be attributed to the influence of T- and

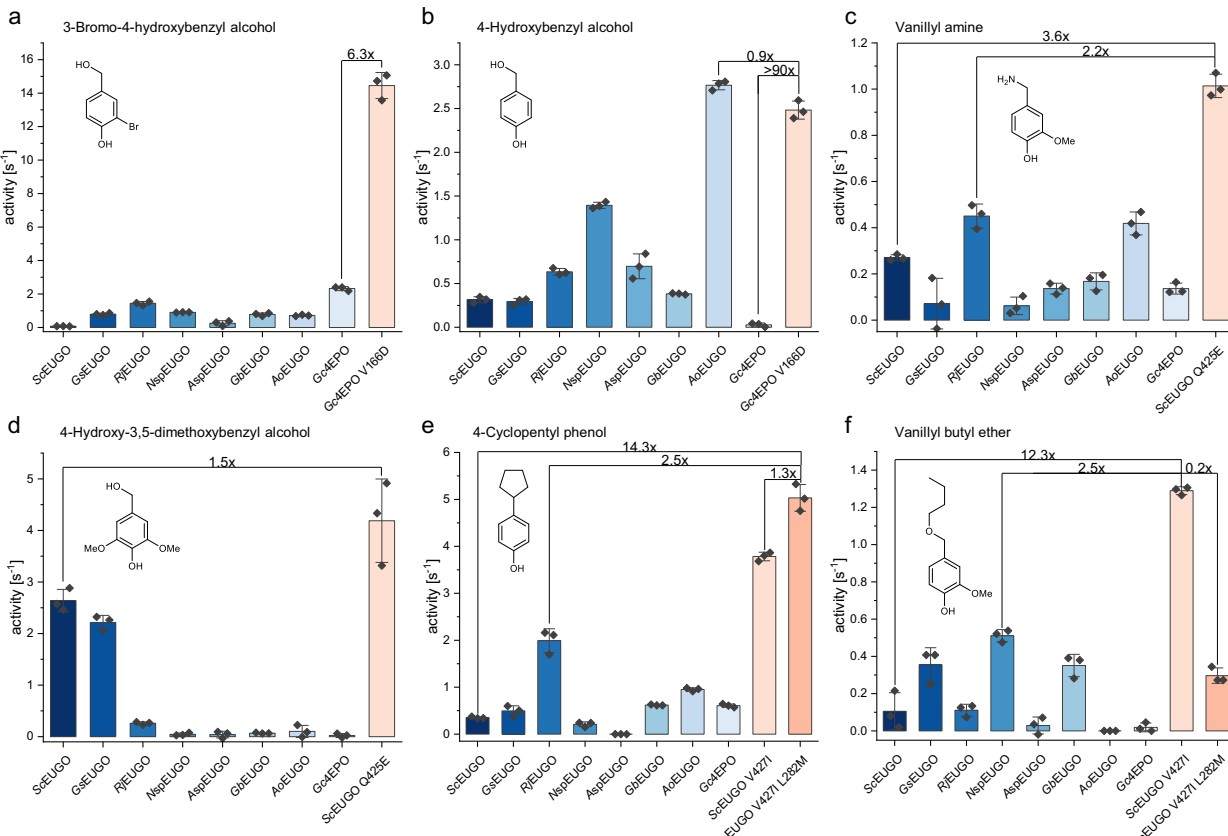

**Fig. 5 | Comparison of the best-performing mutants (orange) with the eight wildtype oxidases (blue) of this study.** The respective improvement to the best-performing enzyme and the wild type is highlighted. The mutants were obtained after screening from a site-saturation mutagenesis library. Rates were determined by xylenol orange assay by applying 2 mM of the substrate in 50 mM potassium phosphate buffer at pH 7.5. The error bars represent the standard deviation from a triplicate measurement. **a** The performance of $Gc$4EPO could be 6.3-fold increased by a V166D mutation for 3-bromo-4-hydroxybenzyl alcohol (**8**), reaching the highest measured rate of $14.5 \pm 0.5$ s$^{-1}$. **b** Activity was restored for $Gc$4EPO on 4-hydroxybenzyl alcohol (**4**) by the V166D mutation. The variant is more than 90 times faster than the wild type and only lags behind AoEUGO. **c** The deamination of

vanillyl amine (**16**) was performed 2.2 times faster by $Sc$EUGO Q425E compared to the best-performing natural enzyme variant, which represents a 3.6-fold improvement compared to the $Sc$EUGO wildtype. **d** $Sc$EUGO Q425E variant was 1.5-fold faster than the respective wildtype on 4-hydroxy-3,5-dimethoxybenzyl alcohol (**6**). **e** The $Sc$EUGO V427I L282M double variant acts 14.3 times faster on 4-cyclopentyl phenol (**42**) and is 2.5-fold faster than the best-performing wild-type enzyme. The introduction of the second mutation increased the rate 1.3-fold. **f** On vanillyl butyl ether (**13**), the $Sc$EUGO V427I variant is 4.3 times faster than the double variant. This represents a 12.3-fold improvement compared to the wildtype and is a 2.5-fold improvement to the best-performing wildtype enzyme.

H-cluster residues for benzyl alcohol derivatives as model compounds (Fig. S26). For 3-bromo-4-hydroxybenzyl alcohol (**8**), the size and polarity of T-cluster residues were identified as determinants for selectivity, while for 3,4-dihydroxybenzyl alcohol (**7**), the size of the residues in positions 381 and 392 are most influential. This is supported by the dominating steric influence found for Ser392 in $As$pEUGO (see above). Overall, the halogen substituent seems to interact with the T-cluster, while the $o$-hydroxy and -methoxy groups rather interact with the H-cluster. In agreement with that, the methoxy group of vanillyl alcohol (**5**) is directed towards the H-cluster in the crystal structure of $Rj$EUGO (Fig. 2a).

The W-cluster was proposed to have the largest influence on the reaction type (H3). To verify this hypothesis, the conversion of five mono-$o$-methoxylated substrates was compared (Fig. S27), for which different reactions are observed: alcohol oxidation (vanillyl alcohol, **5**), deamination (vanillyl amine, **16**), ether cleavage (vanillyl ethyl ether, **12**), dehydrogenation or 4α-hydroxylation (4-ethylguiacol, **34**), and 4γ-hydroxylation (eugenol, **2**). A strong influence of the A-cluster and residue 392 were visible. Further, a strong polar effect in position 282 was observed for **12**. Here, non-polar residues appeared to be beneficial in this position which is opposite to **16**. Thus, ether cleavage and deamination reactions require different polar interactions from the A-cluster. For T-cluster residues, an increasing negative correlation was observed with

regard to the chain length of the $p$-substituent. It can be speculated if increasing repulsion within the A-cluster pushes the substrate molecule towards the tunnel where smaller residues are beneficial to allow an alternative orientation. Along this line, similar patterns are observed for **2** and **12**, which is likely caused by the chain length of the $p$-substituent.

As most selected enzymes are not active on substrates with a chain length larger than three atoms, it was difficult to select suitable substrates to determine factors restricting the activity with regard to the size of the $p$-substituents (H4). For $p$-alkyl substrates, $Gc$4EPO was found the most active enzyme, which strongly biases the analysis (Fig. S28). Diverse activity patterns were only observed for vanillyl butyl ether (**13**) and 4-cyclopentyl phenol (**42**). While for the latter no strong correlations were observed, negative correlations were found for several residues in the A-cluster for **13**, supporting H4. Notably, the negative correlation for residue 392 in the H-cluster may indicate an orientation of the $o$-methoxy group towards this cluster.

## Catalyst enhancement by site-saturation mutagenesis of hot spot residues

To further investigate H3 and H4, and to utilize the obtained structure-function relations to expand the substrate scope of 4-phenol oxidases, site-saturation mutagenesis was performed (Table S7). For this, a peroxidase-

independent screening approach had to be developed (Figs. S29 and S30), as common peroxidases, like horse radish peroxidase, react with the phenolic substrates in a side reaction[37,38]. Building upon the initial hypotheses, three mutation aims were set: (i) Non-natural substituents in *o*-position, (ii) alteration of the W-cluster for improved deamination and ether cleavages, and (iii) sterically demanding groups in *p*-position. *Sc*EUGO was selected as an initial starting point as the wide catalytic pocket was considered most tolerant for residue changes.

For target (i), screening was conducted for A166X and V427X libraries (*Sc*EUGO numbering), resulting in five hits (Table S8). After follow-up investigations in crude extract (Figs. S31 and S32), *Sc*EUGO V427Y was selected as the best candidate, for which a 4.6-fold increase in initial rate ($0.36 \pm 0.02$ s$^{-1}$) for 3-bromo-4-hydroxybenzyl alcohol (**8**, Table S9) was found, in comparison to the wildtype. Nevertheless, the performance of the variant was considered insufficient as the obtained activity was still only 16% of the activity of *Gc*4EPO wildtype on the same substrate (**8**). Therefore, *Gc*4EPO itself was chosen for site-saturation mutagenesis. A single hit was obtained from the screening of V166X and I432X libraries (*Gc*4EPO numbering). The obtained *Gc*4EPO V166D variant was found to be 6.3-times more active on **8** ($14.7 \pm 0.5$ s$^{-1}$) than wildtype enzyme (Fig. 5a). Moreover, the mutation restored the activity on 4-hydroxybenzyl alcohol (**4**), increasing the initial rate more than 90-fold ($2.4 \pm 0.1$ s$^{-1}$, Fig. 5b). Kinetic studies revealed that the increase in activity is $K_M$-driven as the $K_M$ value of the variant towards **8** is about six times higher than for the wildtype ($19 \pm 1$ µM vs. $134 \pm 17$ µM) (Fig. S33, Table S10). Comparably high $K_M$ values were observed for **4** ($230 \pm 63$ µM) and 3,4-dihydroxybenzyl alcohol (**9**, $106 \pm 33$ µM, Figs. S34 and S35). Docking experiments and molecular dynamics simulations revealed that D166 interacts with the phenolate of the substrate molecule and the residues of the P-cluster, causing a slight rotation (Fig. S36). This altered binding mode may be responsible for the observed changes in $K_M$ values compared to the wild-type enzyme.

To increase the sequence space in the W-cluster, *Sc*EUGO E378T and *Sc*EUGO E378Q variants were generated by QuikChange mutagenesis as the limited information obtained from activity correlation suggested that less polar residues (compared to Glu) might be beneficial for deamination reactions. While a strong reduction in 4ɣ-hydroxylation was observed for both variants, minor improvements for 4α-hydroxylation were detected compared to the wild type (Table S9). Both variants reached about 0.2 s$^{-1}$ on 4-ethylguaiacol (**34**), which represents the second highest rate after the outstanding *Gc*4EPO ($2.4 \pm 0.07$ s$^{-1}$). However, the activity for deamination on vanillyl amine (**16**) was reduced by about 50%. Thus, site-saturation mutagenesis was performed for position 425 in a second approach. *Sc*EUGO Q425E was yielded as a single hit, which was found to be 2.2 times more active for the deamination of **16** than the best-performing wildtype enzymes, *Rj*EUGO and *Ao*EUGO, remarking a 3.6-fold improvement from the *Sc*EUGO wildtype (Fig. 5c). In the structural model, the newly introduced carboxyl group of E425 interacts with the amine group of the substrate, leading to a favorable positioning of the benzylic hydrogens (Fig. S37). This good substrate fit of vanillyl amine is in agreement with an observed $K_M$ value of $114 \pm 13$ µM (Fig. S38). In addition to this, a 1.5-fold faster rate for 4-hydroxy-3,5-dimethoxybenzyl alcohol (**6**, $4.1 \pm 0.6$ s$^{-1}$) was detected, compared to *Sc*EUGO wildtype (Fig. 5d). On the downside, the activity for 4ɣ-hydroxylations was reduced by 90% for eugenol (**2**) and 70% for 4-allyl-2,6-dimethoxy phenol (**3**), while no activity was observed at all for 4α-hydroxylations (Table S9). It can be hypothesized that the carboxyl group of E425 interacts with the nucleophilic water, which inhibits the hydroxylation of the substrate molecule.

Before the A-cluster was targeted by mutagenesis according to aim (iii), the catalytic cavity of *Sc*EUGO had to be narrowed down in a first attempt to accommodate substrates without *o*-substituents as substrates with sterically demanding *p*-substituents also contain no substituents in *o*-position. The demand for this strategy was highlighted by an initial screening round on these compounds targeting the A-cluster, which resulted in *Sc*EUGO L381I as a single hit, for which no significant improvement compared to the wild type was detected (Fig. S39).

Since residues in the T-cluster were identified earlier as interaction sites for *o*-substituents (H2), the libraries A166X and V427X were screened, resulting in three hits (Table S8), of which *Sc*EUGO V427I was identified as the most versatile variant (Fig. S40). Notably, the variant was found to have the fastest initial rate on eugenol (**2**) ever reported for an EUGO at neutral pH ($7.2 \pm 0.3$ s$^{-1}$). Regarding substrates with sterically demanding *p*-substituents, 4-cyclohexyl phenol (**42**) and vanillyl butyl ether (**13**), *Sc*EUGO V427I converted both 1.9 and 2.5-times faster than the respective best-performing wildtype enzyme. Compared to *Sc*EUGO wildtype, a 9.5- and 12.3-fold improvement was achieved, respectively (Fig. 5e, f). Thus, the variant was used as a starting point for the second site-saturation mutagenesis round to shape the A-cluster for larger *p*-substituents. From the correlation data, position 282 was suggested as a target since strong correlations were observed for **13** and 4-propyl phenol (**37**, Fig. S28). *Sc*EUGO V427I L282M was obtained as a single hit from library screening, adding a 1.3-fold improvement in the activity on **42** (Table S9), which remarks a 2.5-fold improvement compared to *Rj*EUGO as the best-performing wildtype enzyme and a 14.3-fold improvement compared to *Sc*EUGO wildtype (Fig. 5e, f). A $K_M$ value of $74 \pm 4$ µM was determined, which indicates a good substrate fit (Fig. S41). Interestingly, the activity for **13** decreased in comparison to the single variant. Together with the fact that Met is not considerably smaller than Leu, it becomes clear that other effects are responsible for the increase in activity than steric factors. After docking and simulation experiments, the cyclopentyl ring is positioned similarly for the *Sc*EUGO wildtype and the two variants (Fig. S42). Thus, the higher flexibility of the Met may rather allow for an easier planarization of the ring in the methide intermediate (Fig. S43).

Interestingly, *Sc*EUGO V427I L282M reassembles the catalytic pocket of *Rj*EUGO which was the most active wildtype enzyme for 4-cyclopentyl phenol (Fig. S44). Moreover, the catalytic pocket of the *Sc*EUGO V427I variant is similar to the one from *Nsp*EUGO which is the most active wildtype enzyme on vanillyl butyl ether (Fig. S45). Thus, observed mutations mirror tendencies in the wild-type enzymes. This represents an important confirmation of to the correlations of residues in the catalytic center with activities for the wild-type enzymes. Further, this observation highlights that the catalytic ability for non-natural substrates is disclosed in the sequence space of the natural enzymes, which underlines the successful logic of the presented approach.

## Discussion

In this work, a systematic investigation of the first shell residues in the catalytic center of flavin-dependent 4-phenol oxidoreductases of the VAO/PCMH superfamily was performed. The identification of functional clusters led to the methodical analysis of the enzyme family in the phylogenetic context, which resulted in the identification of bacterial 4-phenol oxidases as a comparably versatile and diverse enzyme family. Thus, this group was chosen as a model system to demonstrate the feasibility of our approach of rational selection of enzymes with novel characteristics based on the chemical properties of the before-defined amino acid clusters. This strategy was enabled by the computational tool A$^2$CA presented in this work, as it connects information from the multiple sequence alignment with the respective phylogenetic data. In total, eight 4-phenol oxidases, of which five were uncharacterized so far, were selected to study the individual influence of every first shell residue on substrate acceptance. Correlations of the residues' properties with the logarithm of the activity of the natural enzymes allowed the conclusion of sequence-function relations for the acceptance of substrates with variable numbers and types of residues in *o*-position as well as for the performed reaction type and the size of the residue in *p*-position. The correlation patterns were supported by kinetic data and structural models to validate hypotheses drawn from literature and propose new theories for previously not investigated residues, like e.g., position 378 (*Rj*EUGO numbering). This fundamental understanding of the sequence–function relations in bacterial 4-phenol oxidases allowed the identification of hot spot residues for subsequent mutagenesis studies. To test hundreds of enzyme variants, a reliable and fast oxidase screening was developed that does not rely on the secondary reaction of a peroxidase,

**Article**

which would interfere with the phenolic substrates of the reaction itself. Site-saturation mutagenesis resulted in sixteen active protein variants and the successful expansion of the substrate scope toward compounds with halogen atoms in the *o*-position, sterically demanding groups in the *p*-position, and improved deamination reactions. Five of the obtained variants performed better than all-natural enzymes, while activity improvements of up to 90 times were achieved with respect to the respective wildtype enzyme. The newly introduced amino acids amplified tendencies observed for the natural enzymes and connect well to the correlation studies, which, overall, underlines the successful logic of the presented approach.

In conclusion, we present a time and resource efficient workflow to disclose the natural sequence space of any enzyme family and to leverage this knowledge to expand it towards non-natural substrates. We demonstrated the approach for the family of bacterial 4-phenol oxidases and hope that the overall concept will be expanded to enzyme families as well.

## Methods

### A2CA analysis
See Supplementary Note 1.

### Heterologous protein production
See Supplementary Methods 1, Supplementary Table 1, and Supplementary Figs. 5–12.

### Sequence analysis
See Supplementary Methods 2 and Supplementary Table 2.

### Thermal stability and substrate conversion
See Supplementary Methods 3 and Supplementary Tables 3–5.

### Homology modeling and molecular dynamics simulations
See Supplementary Methods 4.

### Buffer optimization and pH screening
See Supplementary Methods 5.

### Michaelis-Menten kinetics
See Supplementary Methods 6. Kinetic models used for fitting are provided in Supplementary Eqs. 1 to 5.

### Correlation analysis
See Supplementary Methods 7. The formula used for calculation of the respective correlation coefficients is provided in Supplementary Eq. 6.

### Oxidase screening
See Supplementary Note 2, Supplementary Methods 8, and Supplementary Figs. 29 and 30. Formulas used for normalization during the screening are provided in Supplementary Eqs. 7 to 9.

### Site-directed mutagenesis and characterization of enzyme variants
See Supplementary Methods 8 and Supplementary Tables 7 and 8.

## Data availability
Primary data for phylogenetic analysis and for sequence-activity correlations are provided in Supplementary Data 1 and Supplementary Data 2, respectively. Primary data for point diagrams and calibration rows are compiled in Supplementary Data 4. Raw data of the thermal shift assay is provided in Supplementary Data 5. Further data is included in the Supplementary Information and is available from the authors upon request.

## Code availability
The computational tool A²CA is deposited in the Science Data Bank[34]. The R code for calculation of the correlation patterns is available in Supplementary Data 3.

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

## Acknowledgements
D. Eggerichs was funded by the Deutsche Bundesstiftung Umwelt (20019/625) and was supported by the German Research Council (DFG) within the framework of GRK 2341 (Microbial Substrate Conversion), which was awarded to D. Tischler. The authors thank Prof. M. Fraaije (Groningen, The Netherlands) for provision of the pBADNk vector encoding for *Rj*EUGO.

## Author contributions
D. Eggerichs: Conceptualization, methodology, programming of A$^2$CA, structural analysis, analysis of correlation data, data curation, writing—original draft. N. Weindorf: Calculation of homology models, autodocking, structural analysis, enzyme activity assays, writing. H.G. Weddeling: Site-saturation mutagenesis, development of oxidase screening, enzyme activity assays, Michaelis–Menten kinetics, structural analysis. I.M. Van der Linden: Enzyme characterization, Michaelis–Menten kinetics. D. Tischler: Conceptualization, methodology, supervision, funding acquisition, writing

## Funding

## Competing interests
The authors declare no competing interests.
