## [Peer Review File · Communications Chemistry]

Reviewers' comments:

Reviewer #1 (Remarks to the Author):

In this manuscript, authors use the information in the database to classified the phenol oxidoreductases family. Based the known structures of these enzymes, the cluster of amino acid residue within the active site were classified. Using the obtained bioinformatic data and the comprehensive review, eight oxidases were selected to study. The activity of all expressed oxidases was tested with various substrate scope. The quantitative structure-activity relationship was identified to related the important of amino acid residues toward the catalysis of various types of substrates. According to the systematic study, authors use the obtained information for the enzyme engineering of some selected enzyme by semi-rational enzyme engineering. Some mutants were obtained with the higher activity with the unfamiliar substrate rather than its wildtype and also other natural enzymes. This work provides the comprehensive data collection and data analysis to get the fundamental knowledge about structure and function of phenol oxidase family.

There are some comments from this reviewer:

1. Based on the eight selected oxidases, author should provide the multiple sequence alignment and amino acid sequence similarity of these enzymes. For their relationship, is it the highlight in phylogenetic tree in Figure 3?
2. All enzymes were successfully expressed in *E. coli*. Could author provide the expression profile of these enzymes, for example, some SDS-PAGE analysis? Are these enzymes containing FAD as prosthetic group?
3. Page 6 line 147: Product formation was validated by GC-MS measurements. It would be better to provide the product obtain from the catalysis.
4. Page 8 line 166: Why Glycine in position 392 was selected as the important point of concern over other residues?
5. In Table S2, this reviewer suggests to highlight or emphasize the enzyme/compound couple that gain high activity. Could these data present as a graph? It would be better to see the dominant of each enzyme/compound couple.

6. Page 8 line 170: “For Gc4EPO, no activity was found for chavicol (1) or 4-hydroxybenzyl alcohol (4), while an outstanding activity for 4-ethyl phenol (32) was observed ($4.58 \pm 0.18 \text{ s}^{-1}$, Table S2). This reviewer author should discuss how the Phenylalanine in the position 392 affect to this result? This reviewer also noticed the higher activity was found in Gc4EPO in the catalysis of compound (5) and (7). Are there any meaning for these two compounds?

7. Authors engineer an enzyme based on semi-rational design enzyme engineering to obtained the efficient mutants. However, some of the substrates were tested. Could the author provide or discuss the significant of tested substrate and the obtained products from the substrate expansion?

Reviewer #2 (Remarks to the Author):

The Eggerichs et al. made an excellent engineering endeavor in the quest to expand substrate scope of 4-Phenol Oxidase. Guided by the computational analysis, the appropriate mutational residues (hotspots) were identified. Following the subsequent site-directed mutagenesis, a set of single-site variants with improved activity toward distinct substrates were successfully obtained. Overall, the story is interesting, and the manuscript is well organized. My major comment is that performing combinatorial mutagenesis on the beneficial single-point substitutions to explore the potential epistasis effects would be necessary, and may further improve the quality of the manuscript.

The other minor comment is that the authors should check the language extensively. For examples, in line 11, catalyst should be catalysts. In line 262, side-saturation should be site-saturation.

Answer to the reviewers:

Reviewer #1

1. Based on the eight selected oxidases, author should provide the multiple sequence alignment and amino acid sequence similarity of these enzymes. For their relationship, is it the highlight in phylogenetic tree in Figure 3?

A respective sequence alignment was added to the additional information and is mentioned in the main text in line 141. Further, table S9 was included, which states the sequence identity of all sequences towards the best characterized enzyme form *Rhodococcus jostii* RHA1.

Yes, the enzymes of this study are color coded in the phylogenetic tree in figure 3A. In literature characterized enzymes (light red), in literature characterized enzymes used in this study (light orange) and enzymes characterized in this study (green) are differentiated.

2. All enzymes were successfully expressed in *E. coli*. Could author provide the expression profile of these enzymes, for example, some SDS-PAGE analysis? Are these enzymes containing FAD as prosthetic group?

The expression profile and the absorption spectrum of all eight oxidases were added to the additional information. The UV/vis spectrum of the enzymes show a peak at 441 nm, which corresponds to the well-studied *RjEUGO* as characteristic peak of the covalently bound FAD as prosthetic group. In addition, we added a phrase in the introduction to reference to previously studied enzymes (line 65-66).

3. Page 6 line 147: Product formation was validated by GC-MS measurements. It would be better to provide the product obtain from the catalysis.

Table S3 in the additional information now provides an overview over substrates and products detected.

4. Page 8 line 166: Why Glycine in position 392 was selected as the important point of concern over other residues?

The paragraph was rephrased to clarify this point. Briefly, the position stood out as most variable position and was therefore investigated in more detail.

5. In Table S2, this reviewer suggests to highlight or emphasize the enzyme/compound couple that gain high activity. Could these data present as a graph? It would be better to see the dominant of each enzyme/compound couple.

The cells in table S2 and S6 were colored according to their value from highest (blue) to lowest (light orange) rate. We hope that this representation gives the reader a better overview of the differences in initial rates observed.

6. Page 8 line 170: "For Gc4EPO, no activity was found for chavicol (1) or 4-hydroxybenzyl alcohol (4), while an outstanding activity for 4-ethyl phenol (32) was observed (4.58 ± 0.18 s⁻¹, Table S2). This reviewer author should discuss how the Phenylalanine in the position 392 affect to this result? This reviewer also noticed the higher activity was found in Gc4EPO in the catalysis of compound (5) and (7). Are there any meaning for these two compounds?

The suggestions were included in the section "*Modulation of enzyme activity through substrate rotation by residue 392*". With regard to the high activities of compound (5) and (7), it is likely caused

by the introduction of larger residues in the A-cluster as it was described by Alvigini et al. These residues result in a better fit for benzyl alcohols.

7. Authors engineer an enzyme based on semi-rational design enzyme engineering to obtain the efficient mutants. However, some of the substrates were tested. Could the author provide or discuss the significance of tested substrate and the obtained products from the substrate expansion?

In general, the aim of the study is to demonstrate the rationalization of the catalytic pocket in case of 4-phenol oxidases. Thus, the substrates were selected to investigate the hypotheses H1 to H4. For the wildtype enzymes, all substrates were measured to obtain a database for detailed analysis. For the generated mutants, only substrates were tested which are relevant for the hypotheses H1 to H4 and the mutagenesis goals (i) to (iii).

Reviewer #2

The Eggerichs et al. made an excellent engineering endeavor in the quest to expand substrate scope of 4-Phenol Oxidase. Guided by the computational analysis, the appropriate mutational residues (hotspots) were identified. Following the subsequent site-directed mutagenesis, a set of single-site variants with improved activity toward distinct substrates were successfully obtained. Overall, the story is interesting, and the manuscript is well organized. My major comment is that performing combinatorial mutagenesis on the beneficial single-point substitutions to explore the potential epistasis effects would be necessary and may further improve the quality of the manuscript. The other minor comment is that the authors should check the language extensively. For examples, in line 11, catalyst should be catalyts. In line 262, side-saturation should be site-saturation.

The key objective of the present study was the identification of hot spot residues in the family of 4-phenol oxidases to demonstrate the streamlined selection of appropriate biocatalysts from a fast sequence space. The mutagenesis experiments were used to validate the hypotheses H1 to H4 about the catalytic center. We agree that combinatorial mutagenesis would be very interesting but should in our opinion be part of a different story. For the two double variants which were created during this study, mixed results were observed. On the one hand, ScEUGO D151N Q425E was created in expectation of an epistasis effect, but the activity of the enzyme could not be restored. On the other hand, the introduction of the L282M mutation into the ScEUGO V427I variant increased the overall activity by another 30%. Thus, in this case, an additive effect of the second mutation could be observed. The language of the manuscript was checked again. We apologize for any inconvenience during the revision process.

REVIEWERS' COMMENTS:

Reviewer #1 (Remarks to the Author):

The Authors have carefully and comprehensively addressed the Reviewers' comments. They address the relationship between amino-acid sequences/Structure with the function using the many substrates. The manuscript will be a relevant contribution and add to our knowledge on Flavin-dependent 4-Phenol oxidase.